# PatchDCT: Patch Refinement for High Quality Instance Segmentation

**Qinrou Wen[1], Jirui Yang[2], Xue Yang[3], Kewei Liang[1,*]**
[1]School of Mathematical Sciences, Zhejiang University    [2]Alibaba Group
[3]Department of CSE, MoE Key Lab of Artificial Intelligence, Shanghai Jiao Tong University
{qinrou.wen,matlkw}@zju.edu.cn,    jirui.yjr@alibaba-inc.com
yangxue-2019-sjtu@sjtu.edu.cn
PyTorch Code: https://github.com/olivia-w12/PatchDCT

## Abstract

High-quality instance segmentation has shown emerging importance in computer vision. Without any refinement, DCT-Mask directly generates high-resolution masks by compressed vectors. To further refine masks obtained by compressed vectors, we propose for the first time a compressed vector based multi-stage refinement framework. However, the vanilla combination does not bring significant gains, because changes in some elements of the DCT vector will affect the prediction of the entire mask. Thus, we propose a simple and novel method named PatchDCT, which separates the mask decoded from a DCT vector into several patches and refines each patch by the designed classifier and regressor. Specifically, the classifier is used to distinguish mixed patches from all patches, and to correct previously mispredicted foreground and background patches. In contrast, the regressor is used for DCT vector prediction of mixed patches, further refining the segmentation quality at boundary locations. Experiments on COCO show that our method achieves 2.0%, 3.2%, 4.5% AP and 3.4%, 5.3%, 7.0% Boundary AP improvements over Mask-RCNN on COCO, LVIS, and Cityscapes, respectively. It also surpasses DCT-Mask by 0.7%, 1.1%, 1.3% AP and 0.9%, 1.7%, 4.2% Boundary AP on COCO, LVIS and Cityscapes. Besides, the performance of PatchDCT is also competitive with other state-of-the-art methods.

## 1 Introduction

Instance segmentation (Li et al., 2017; He et al., 2017) is a fundamental but challenging task in computer vision, which aims to locate objects in images and precisely segment each instance. The mainstream instance segmentation methods follow Mask-RCNN (He et al., 2017) paradigm, which often segment instances in a low-resolution grid (Kang et al., 2020; Cheng et al., 2020c; Chen et al., 2019; Ke et al., 2021). However, limited by the coarse mask representation ( i.e. $28 \times 28$ in Mask-RCNN), most of these algorithms cannot obtain high-quality segmentation results due to the loss of details. DCT-Mask (Shen et al., 2021) achieves considerable performance gain by predicting an informative 300-dimensional Discrete Cosine Transform (DCT) (Ahmed et al., 1974) vector compressed from a $128 \times 128$ mask. To further improve the segmentation results of DCT-Mask, we follow the refine mechanism (Ke et al., 2022; Zhang et al., 2021; Kirillov et al., 2020) to correct the mask details in a multi-stage manner.

A straightforward implementation is to refine the 300-dimensional DCT vector multiple times. However, experimental results show that this naive implementation does not succeed, which improves mask average precision (mAP) by 0.1% from 36.5% to 36.6% on COCO *val set*. The main reason for the limited improvement is that the full 300-dimensional DCT vector is not suitable for refining some important local regions, such as wrong predicted regions and boundary regions in masks. As each pixel value in the mask is calculated by all elements of the DCT vector in the inference stage, once some elements in the DCT vector change, the entire mask will change, and even the correct segmentation areas may be affected, refer to Figure 1a.

---

*Corresponding author is Kewei Liang.

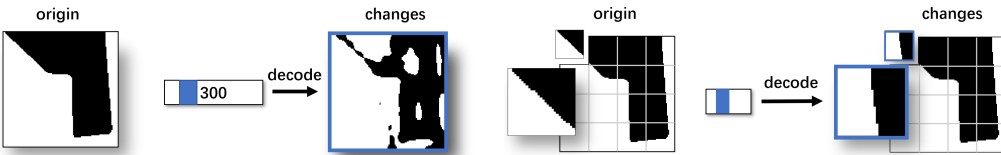

(a) Influence of element change in DCT-Mask       (b) Influence of element change in PatchDCT

Figure 1: (a) Influence of elements changes in DCT vectors for DCT-Mask. The blue block denotes the changed elements. The box with a blue border represents the part of the mask affected by the changes in element values. The change of some elements will affect the entire mask. (b) Influence of elements changes in DCT vectors for PatchDCT. Changing some elements of a vector will only affect the corresponding patch.

To overcome the above issue, we propose a novel method, called PatchDCT, which divides the mask decoded from a DCT vector into several independent patches and refines each patch with a three-class classifier and a regressor, respectively. In detail, each patch is first classified into one of three categories: foreground, background, and mixed by the classifier, and then previously mispredicted foreground and background patches will be corrected. Mixed patches are fed into the regressor to predict their corresponding $n$-dimensional ($n \ll 300$) DCT vectors. In the inference stage, we use Inverse Discrete Cosine Transform (IDCT) to decode the predicted vectors of the mixed patches as their refined masks, and merge them with the masks of other foreground and background patches to obtain a high-resolution mask. It is also worth emphasizing that each patch is independent, so the element change of a DCT vector will only affect the corresponding mixed patch, as shown in Figure 1b. In general, patching allows the model to focus on the refinement of local regions, thereby continuously improving the quality of segmentation, resulting in significant performance improvements. Our main contributions are:

**1)** To our best knowledge, PatchDCT is the first compressed vector based multi-stage refinement detector to predict high-quality masks.

**2)** PatchDCT innovatively adopts the patching technique, which successfully allows the model to focus on the refinement of important local regions, fully exploiting the advantages of multi-stage refinement and high-resolution information compression.

**3)** Compared to Mask RCNN, PatchDCT improves about 2.0% AP and 3.4% Boundary AP on COCO, 3.2% AP and 5.3% Boundary AP on LVIS*[1], 4.5% AP and 7.0% Boundary AP on Cityscapes. It also achieves 0.7% AP and 0.9% Boundary AP on COCO, 1.1% AP and 1.7% Boundary AP on LVIS*, 1.3% AP and 4.2% Boundary AP on Cityscapes over DCT-Mask.

**4)** Demonstrated by experiments on COCO *test-dev*, the performance of PatchDCT is also competitive with other state-of-the-art methods.

## 2   RELATED WORK

**Instance segmentation.** Instance segmentation assigns a pixel-level mask to each instance of interest. Mask-RCNN (He et al., 2017) generates bounding boxes for each instance with a powerful detector (Ren et al., 2015) and categorizes each pixel in bounding boxes as foreground or background to obtain $28 \times 28$ binary grid masks. Several methods that build on Mask-RCNN improve the quality of masks. Mask Scoring RCNN (Huang et al., 2019) learns to regress mask IoU to select better-quality instance masks. HTC (Chen et al., 2019) utilizes interleaved execution, mask information flow, and semantic feature fusion to improve Mask-RCNN. BMask RCNN (Cheng et al., 2020c) adds a boundary branch on Mask-RCNN to detect the boundaries of masks. Bounding Shape Mask R-CNN (Kang et al., 2020) improves performance on object detection and instance segmentation by its bounding shape mask branch. BCNet (Ke et al., 2021) uses two GCN (Welling & Kipf, 2016) layers to detect overlapping instances. Although these algorithms have yielded promising results, they are still restricted in the low-resolution mask representation and thus do not generate high-quality masks.

---

[1]COCO dataset with LVIS annotations

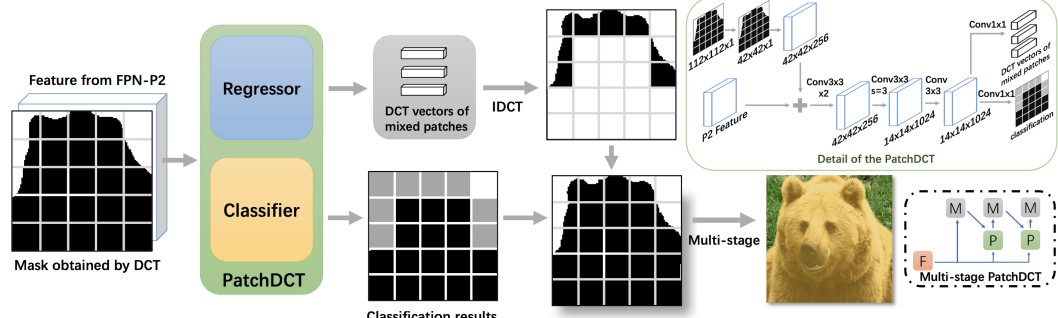

Figure 2: The pipeline of PatchDCT. The classifier differentiates foreground, background and mixed patches. The regressor predicts the DCT vectors of mixed patches. Masks of mixed patches are obtained by patch DCT vectors. PatchDCT combines masks of all patches to obtain an entire mask of instance. The entire mask of instance output by PatchDCT can be fed into another PatchDCT module for a finer mask. For the architecture of multi-stage PatchDCT: 'F' is the feature map cropped from FPN-P2. 'M' is the high-resolution mask. 'P' is the PatchDCT module.

**Towards high-quality instance segmentation.** To take full advantage of high-resolution masks, DCT-Mask (Shen et al., 2021) learns to regress a 300-dimensional DCT vector compressed from a $128 \times 128$ mask. SOLQ (Dong et al., 2021) is a query-based method, which also encodes high-resolution masks into DCT vectors and predicts the vectors by queries. Both of these methods generate high-resolution masks in a one-shot manner, without any refinement. Although they have made considerable gains, there is still potential for improvement. Multi-stage refinement is another common technique for obtaining high-quality masks. PointRend (Kirillov et al., 2020) adaptively selects several locations to refine, rendering $224 \times 224$ masks from $7 \times 7$ coarse masks. RefineMask (Zhang et al., 2021) introduces semantic segmentation masks as auxiliary inputs, and generates $112 \times 112$ masks in a multi-stage manner. Mask Transfiner (Ke et al., 2022) represents image regions as a quadtree and corrects the errors of error-prone tree nodes to generate $112 \times 112$ masks. PBR (Tang et al., 2021) is a post-processing method that refines patches along the mask boundaries. Unlike these refinement methods based on the binary grid mask representation, our method is based on compressed vectors.

Generating high-quality masks is also one of the main concerns in the field of semantic segmentation. CRFasRNN (Zheng et al., 2015) connects CRF (Krähenbühl & Koltun, 2011) with FCN (Long et al., 2015), formulating mean-field approximate inference for the CRF with Gaussian pairwise potentials as Recurrent Neural Networks. DeepLab (Chen et al., 2017) effectively improves the quality of masks by using atrous convolution for receptive field enhancement, ASPP for multi-scale segmentation, and CRF for boundary refinement. SegModel (Shen et al., 2017) utilizes a guidance CRF to improve the segmentation quality. CascadePSP (Cheng et al., 2020b) trains independently a refinement module designed in a cascade fashion. RGR (Dias & Medeiros, 2018) is a post-processing module based on region growing. In contrast, PatchDCT can obtain high-quality segmentation results in an end-to-end learning manner without any additional post-processing.

## 3 METHODS

In this section, we show the difficulties in refining DCT vectors and then introduce PatchDCT to overcome these difficulties and generate finer masks.

### 3.1 DIFFICULTIES IN REFINING DCT VECTORS

Given a $K \times K$ mask, DCT-Mask (Shen et al., 2021) encodes the mask $\mathbf{M}_{K \times K}$ into the frequency domain $\mathbf{M}_{K \times K}^{f}$:

$$M_{K \times K}^{f}(u, v) = \frac{2}{K} C(u) C(v) \sum_{x=0}^{K-1} \sum_{y=0}^{K-1} M_{K \times K}(x, y) \cos \frac{(2x+1)u\pi}{2K} cos \frac{(2y+1)v\pi}{2K}, \quad (1)$$

where $C(w) = 1/\sqrt{2}$ for $w = 0$ and $C(w) = 1$ otherwise. Non-zero values are concentrated in the upper left corner of $\mathbf{M}^f_{K \times K}$, which are low-frequency elements that contain the most information of the mask. The $N$-dimensional DCT vector is obtained by zigzag scanning (Al-Ani & Awad, 2013) $\mathbf{M}^f_{K \times K}$ and selecting the top-$N$ elements.

In the inference stage, $\mathbf{M}^f_{K \times K}$ is recovered by filling the remaining elements to zero. Then each pixel in the mask $\mathbf{M}_{K \times K}$ is calculated as follow:

$$M_{K \times K}(x, y) = \frac{2}{K} C(x) C(y) \sum_{u=0}^{K-1} \sum_{v=0}^{K-1} M^f_{K \times K}(u, v) \cos \frac{(2x+1)u\pi}{2K} cos \frac{(2y+1)v\pi}{2K}, \quad (2)$$

Equation 2 reveals that each pixel in the mask $\mathbf{M}_{K \times K}$ is calculated by all elements of $\mathbf{M}^f_{K \times K}$. When refining the $N$-dimensional DCT vector, once an element is incorrectly changed, all pixels in $\mathbf{M}_{K \times K}$ will be affected, even those correctly segmented regions, which is also shown in Figure 1. Therefore, when fixing some specific error regions (e.g. borders), it is difficult to get the correct refinement result unless all the elements in the DCT vector are correctly refined. In practice, however, it is almost impossible to correctly predict all $N$ elements.

## 3.2 PATCHDCT

To prevent the above issue when refining the global DCT vector, we propose a method named PatchDCT, which divides the $K \times K$ mask into $m \times m$ patches and refines each patch respectively. The overall architecture of PatchDCT is shown in Figure 2, which mainly consists of a three-class classifier and a DCT vector regressor. Specifically, the classifier is used to identify mixed patches and refine foreground and background patches. Each mixed patch is then refined by an $n$-dimensional DCT vector, which is obtained from the DCT vector regressor.

**Three-class classifier.** We define the patches with only foreground pixels and only background pixels as foreground patches and background patches, respectively, while the others are mixed patches. The task of differentiating patch categories is accomplished by a fully convolutional three-class classifier. Moreover, the mispredicted initial foreground and background patches are corrected by the classifier. We utilize a three-class classifier instead of a DCT vector regressor to refine foreground and background patches because of the particular form of their DCT vectors. For background patches, simply from Equation 1, all elements of DCT vectors are zero. For foreground patches, all elements are zero except for the first element named DC component (DCC), which is equal to the patch size $m$. The mathematical proof of the DCT vector form for the foreground patches is shown in the Appendix. DCT vector elements of foreground and background

Table 1: Mask AP obtained by different lengths of ground-truth DCT vectors using Mask-RCNN framework on COCO *val2017*. The $1 \times 1$ patch size represents the binary grid mask representation. Low-dimensional DCT vectors are able to provide enough ground truth information.

| Resolution | Patch Size | Dim. | AP |
|---|---|---|---|
| $112 \times 112$ | $1 \times 1$ | 1 | 57.6 |
| $112 \times 112$ | $8 \times 8$ | 3 | 55.8 |
| $112 \times 112$ | $8 \times 8$ | 6 | 57.1 |
| $112 \times 112$ | $8 \times 8$ | 9 | 57.5 |
| $112 \times 112$ | $8 \times 8$ | 12 | 57.5 |
| $112 \times 112$ | $112 \times 112$ | 200 | 55.8 |
| $112 \times 112$ | $112 \times 112$ | 300 | 56.4 |

patches are discrete data that are more suitable for classification. Referring to Figure 3, DCT vector elements of mixed patches are continuously distributed and therefore more suitable for regression.

**Regressor.** Similar to the phenomenon described in DCT-Mask (Shen et al., 2021), refining high-resolution masks with the binary grid mask representation introduces performance degradation due to the high training complexity (refer to DCT-Mask (Shen et al., 2021) for more details). Learning to regress informative DCT vectors eases the training process. The specific experimental results are discussed in the experiments section (Sec. 4).

The regressor is trained and inferred for mixed patches only. It is actually a boundary attention module, since the mixed patches are distributed exactly along the boundary of the instance mask. For each mixed patch, the regressor predicts an $n$-dimensional DCT vector, which is very short but highly informative. Table 1 shows mask AP obtained by different lengths of ground truth patch DCT

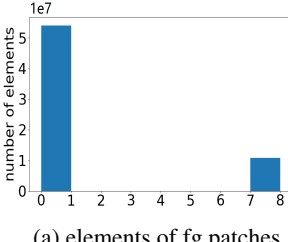 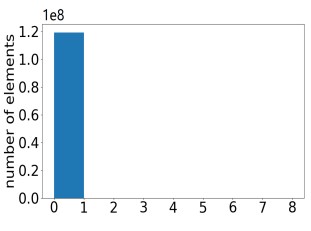 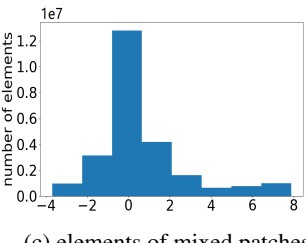

(a) elements of fg patches     (b) elements of bg patches     (c) elements of mixed patches

Figure 3: Elements of 6-dimensional DCT vectors for foreground, background and mixed patches on COCO *val2017*. DCT vector elements for foreground and background patches are discrete data. DCT vector elements for mixed patches are continuous data.

vectors using Mask-RCNN framework on COCO *val2017*. The low-dimensional DCT vectors have been able to provide sufficient ground truth information.

### 3.3 MULTI-STAGE REFINEMENT AND LOSS FUNCTION

PatchDCT is a module where the input and output masks have the same resolution. Thus, the mask generated by a PatchDCT module can be fed into another PatchDCT module for further refinement, as shown in the upper right corner of Figure 2.

With multi-stage refinement, the loss function of the mask branch is defined as

$$\mathcal{L}_{mask} = \lambda_0 \mathcal{L}_{dct_N} + \sum_{s>0} \lambda_s (\mathcal{L}^s_{cls_{patch}} + \mathcal{L}^s_{dct_n}), \tag{3}$$

$\lambda_0$ and $\lambda_s$ are the loss weights. The first item $\mathcal{L}_{dct_N}$ of Equation 3 is the loss in predicting $N$-dimensional vectors of the entire masks (Shen et al., 2021).

$$\mathcal{L}_{dct_N} = \frac{1}{N} \sum_i^N R(\hat{V}_i - V_i), \tag{4}$$

where $V_i$ and $\hat{V}_i$ are the $i$-th element in ground-truth and the prediction vector respectively. $R$ is the loss function and $N$ is the length of the vectors. The classification loss $\mathcal{L}^s_{cls_{patch}}$ of $s$-th stage is the cross-entropy loss over three classes. The regression loss $\mathcal{L}^s_{dct_n}$ of $s$-th stage is

$$\mathcal{L}^s_{dct_n} = \frac{1}{N_m} \sum_k^{N_{all}} \left[ p^k \left( \frac{1}{n} \sum_i^n R(\hat{V}_i - V_i) \right) \right], \tag{5}$$

where $N_m$, $N_{all}$ are the number of mixed patches and all patches respectively. $n$ is the length of the patch DCT vectors. If the $k$-th patch is a mixed patch, $p^k = 1$, otherwise $p^k = 0$, indicating that only DCT vectors of mixed patches are regressed.

## 4 EXPERIMENTS

### 4.1 DATASETS

We evaluate our method on two standard instance segmentation datasets: COCO (Lin et al., 2014) and Cityscapes (Cordts et al., 2016). COCO provides 80 categories with instance-level annotations. Cityscapes is a dataset focused on urban street scenes. It contains 8 categories for instance segmentation, providing 2,975, 500 and 1,525 high-resolution images ($1,024 \times 2,048$) for training, validation, and test respectively.

We report the standard mask AP metric and the Boundary AP (Cheng et al., 2021) metric ($AP_B$), the latter focusing on evaluating the boundary quality. Following (Kirillov et al., 2020), we also report $AP^*$ and $AP^*_B$, which evaluate COCO *val2017* with high-quality annotations provided by LVIS (Gupta et al., 2019). Note that for $AP^*$ and $AP^*_B$, models are still trained on COCO *train2017*.

Table 2: Mask AP on COCO with different backbones based on Mask-RCNN framework. $AP^*$ is results obtained from COCO with LVIS annotations. $AP_B$ is Boundary AP. $AP_B^*$ is Boundary AP using LVIS annotations. Models with R101-FPN and RX101-FPN are trained with '3×' schedule. Runtime is measured on a single A100. Considering the significant improvement of masks, the cost in the runtime is almost negligible.

| Backbone | Model | AP | $AP_S$ | $AP_M$ | $AP_L$ | $AP_B$ | $AP^*$ | $AP_S^*$ | $AP_M^*$ | $AP_L^*$ | $AP_B^*$ | FPS |
|---|---|---|---|---|---|---|---|---|---|---|---|---|
| | Mask-RCNN | 35.2 | 17.2 | 37.7 | 50.3 | 21.1 | 37.6 | 21.3 | 43.7 | 55.1 | 24.8 | **13.9** |
| R50-FPN | DCT-Mask | 36.5 | 17.7 | 38.6 | 51.9 | 23.6 | 39.7 | **23.5** | 46.5 | 58.5 | 28.4 | 13.2 |
| | PatchDCT | **37.2** | **18.3** | **39.5** | **54.2** | **24.5** | **40.8** | 23.0 | **47.7** | **60.7** | **30.1** | 12.3 |
| | Mask-RCNN | 38.6 | 19.5 | 41.3 | 55.3 | 24.5 | 41.4 | 24.5 | 47.9 | 61.0 | 29.0 | **13.8** |
| R101-FPN | DCT-Mask | 39.9 | 20.2 | 42.6 | 57.3 | 26.8 | 43.7 | 25.8 | 50.5 | 64.6 | 32.4 | 13.0 |
| | PatchDCT | **40.5** | **20.8** | **43.3** | **57.7** | **27.6** | **44.4** | **27.0** | **51.5** | **65.3** | **33.8** | 11.8 |
| | Mask-RCNN | 39.5 | 20.7 | 42.0 | 56.5 | 25.3 | 42.1 | 25.4 | 48.0 | 61.4 | 29.7 | **13.3** |
| RX101-FPN | DCT-Mask | 41.2 | 21.9 | 44.2 | 57.7 | 28.0 | 45.2 | 27.4 | 52.6 | 64.2 | 34.0 | 12.9 |
| | PatchDCT | **41.8** | **22.5** | **44.6** | **58.7** | **28.6** | **46.1** | **27.8** | **53.0** | **66.1** | **35.4** | 11.7 |

Table 3: Results on Cityscapes *val* set. $AP_B$ is Boundary AP. All models are based on R50-FPN backbone. PatchDCT achieves the best performance.

| Methods | Resolution | AP | $AP_{50}$ | $AP_B$ |
|---|---|---|---|---|
| Mask-RCNN (He et al., 2017) | $28 \times 28$ | 33.7 | 60.9 | 11.8 |
| Panoptic-DeepLab (Cheng et al., 2020a) | - | 35.3 | 57.9 | 16.5 |
| PointRender (Kirillov et al., 2020) | $224 \times 224$ | 35.9 | 61.8 | 16.7 |
| DCT-Mask (Shen et al., 2021) | $112 \times 112$ | 36.9 | 62.9 | 14.6 |
| RefineMask (Zhang et al., 2021) | $112 \times 112$ | 37.6 | 63.3 | 17.4 |
| Mask Transfiner (Ke et al., 2022) | $112 \times 112$ | 37.9 | 64.1 | 18.0 |
| PatchDCT (Ours) | $112 \times 112$ | **38.2** | **64.5** | **18.8** |

## 4.2 IMPLEMENT DETAILS

We build the model based on DCT-Mask (Shen et al., 2021). We first decode the 300-dimensional DCT vector to obtain a $112 \times 112$ mask. This mask is then fed into PatchDCT, together with a $42 \times 42$ feature map cropped from FPN-P2 (Lin et al., 2017). PatchDCT refines each patch of the mask and outputs a $112 \times 112$ mask. We set the patch size to 8 and each patch is represented by a 6-dimensional DCT vector. Our model is class-specific by default, i.e. one mask per class. $L1$ loss and cross-entropy loss are used for DCT vector regression and patch classification respectively. By default, only one PatchDCT module is used, and both $\lambda_0$ and $\lambda_1$ are set to 1. We implement our algorithm based on Detectron2 (Wu et al., 2019), and all hyperparameters remain the same as Mask-RCNN in Detectron2. Unless otherwise stated, $1\times$ learning schedule is used.

## 4.3 MAIN RESULTS

**Results on COCO.** We compare PatchDCT with Mask-RCNN and DCT-Mask over different backbones. As shown in Table 2, on COCO *val2017* with R50-FPN, PatchDCT improves 2.0% AP and 3.4% $AP_B$ over Mask-RCNN. Compared with DCT-Mask, PatchDCT also achieves 0.7% AP and 0.9% $AP_B$ improvements. When evaluating with LVIS annotations, PatchDCT yields significant gains of 3.2% $AP^*$ and 5.3% $AP_B^*$ over Mask-RCNN, and 1.1% $AP^*$ and 1.7% $AP_B^*$ over DCT-Mask. Consistent improvements are observed on R101-FPN and RX101-FPN. Since $AP^*$ and $AP_B^*$ are evaluated with high-quality annotations, the significant improvements of these two metrics emphasize the superiority of our model. In addition, considering the improvement in mask quality, the cost in runtime is almost negligible, i.e. about 1.5 FPS degradation on the A100 GPU.

We also compare the performance of PatchDCT with state-of-the-art methods of instance segmentation on COCO *test-dev2017*. With RX101 backbone, PatchDCT surpasses PointRender (Kirillov et al., 2020) and RefineMask (Zhang et al., 2021), which are both multi-stage refinement methods based on binary grid masks, by 0.8% and 0.4%. PatchDCT also achieves comparable performance with Mask Transfiner (Ke et al., 2022) with R101 backbone. However, Mask-Transfer runs at 5.5 FPS on the A100 GPU, which is almost two times slower than PatchDCT. With Swin-B back-

Table 4: Comparison of different methods on COCO *test-dev2017*. MS denotes using multi-scale training. '3×' schedules indicates 36 epochs for training. Runtime is measured on a single A100.

| Method | Backbone | MS | Sched. | AP | $AP_{50}$ | $AP_{75}$ | $AP_S$ | $AP_M$ | $AP_L$ | FPS |
|---|---|---|---|---|---|---|---|---|---|---|
| BMask RCNN (Cheng et al., 2020c) | R101-FPN | | 1× | 37.7 | 59.3 | 40.6 | 16.8 | 39.9 | 54.6 | - |
| Mask-RCNN (He et al., 2017) | R101-FPN | ✓ | 3× | 38.8 | 60.9 | 41.9 | 21.8 | 41.4 | 50.5 | 13.8 |
| BCNet (Ke et al., 2021) | R101-FPN | ✓ | 3× | 39.8 | 61.5 | 43.1 | 22.7 | 42.4 | 51.1 | - |
| DCT-Mask (Shen et al., 2021) | R101-FPN | ✓ | 3× | 40.1 | 61.2 | 43.6 | 22.7 | 42.7 | 51.8 | 13.0 |
| Mask Transfiner (Ke et al., 2022) | R101-FPN | ✓ | 3× | 40.7 | - | - | 23.1 | 42.8 | 53.8 | 5.5 |
| SOLQ (Dong et al., 2021) | R101-FPN | ✓ | 50e | 40.9 | - | - | 22.5 | 43.8 | 54.6 | 10.7 |
| MEInst (Zhang et al., 2020) | RX101-FPN | ✓ | 3× | 36.4 | 60.0 | 38.3 | 21.3 | 38.8 | 45.7 | - |
| HTC (Chen et al., 2019) | RX101-FPN | | 20e | 41.2 | 63.9 | 44.7 | 22.8 | 43.9 | 54.6 | 4.3 |
| PointRend (Kirillov et al., 2020) | RX101-FPN | ✓ | 3× | 41.4 | 63.3 | 44.8 | 24.2 | 43.9 | 53.2 | 8.4 |
| RefineMask (Zhang et al., 2021) | RX101-FPN | ✓ | 3× | 41.8 | - | - | - | - | - | 8.9 |
| Mask Transfiner (Ke et al., 2022) | Swin-B | ✓ | 3× | 45.9 | 69.3 | 50.0 | 28.7 | 48.3 | 59.4 | 3.5 |
| PatchDCT (Ours) | R101-FPN | ✓ | 3× | 40.7 | 61.8 | 44.2 | 22.8 | 43.2 | 52.8 | 11.8 |
| PatchDCT (Ours) | RX101-FPN | ✓ | 3× | 42.2 | 64.0 | 45.8 | 25.0 | 44.5 | 53.9 | 11.7 |
| PatchDCT (Ours) | Swin-B | ✓ | 3× | **46.6** | **69.7** | **50.8** | **29.0** | **49.0** | **59.9** | 7.3 |

bone, PatchDCT outperforms Mask Transfiner (Ke et al., 2022) by 0.7% AP. It is worth noting that PatchDCT is faster than most multi-stage refinement methods since only one refine process is required. These results demonstrate the effectiveness of PatchDCT in generating high-quality masks.

**Results on Cityscapes.** We also report results on Cityscapes *val* set in Table 3. In comparison with Mask-RCNN, PatchDCT obtains 4.5% AP and 7.0% $AP_B$ improvements. It also outperforms DCT-Mask by 1.3% AP and 4.2% $AP_B$. Compared with other SOTA methods, PatchDCT is still competitive. PatchDCT achieves 0.8%, 1.4%, 2.1% $AP_B$ gains over Mask Transfiner (Ke et al., 2022), RefineMask (Zhang et al., 2021) and PointRender (Kirillov et al., 2020) respectively. The large difference in $AP_B$ highlights the ability of PatchDCT to generate masks with more detailed borders.

## 4.4 Ablation Experiments

We conduct extensive ablation experiments to further analyze PatchDCT. We adopt R50-FPN as the backbone and evaluate the performance on COCO *val2017*.

**Simply refine DCT vectors.** Simply refining the global DCT vectors does not succeed. To demonstrate that, we design a model named 'Two-stage DCT', which regresses a new 300-dimensional DCT vector after fusing the initial mask with a $42 \times 42$ feature map from FPN-P2. The refined mask is decoded from the final DCT vector. From Table 5, Two-stage DCT achieves only little improvements over DCT-Mask, since changes in some elements of the global DCT vector may affect the entire mask, even for the correct segmentation areas. PatchDCT leverages the patching mechanism to overcome this issue and outperforms Two-stage DCT by 1.0 $AP_B^*$.

**Binary grid refinement.** Refining masks with the binary grid mask representation can be considered as the extreme patching mechanism, which treats each pixel as a patch. However, simply refining high-resolution masks with the binary grid mask representation introduces performance degradation. We construct an experiment named 'binary grid refinement', which predicts another $112 \times 112$ mask with the binary grid mask representation after fusing the initial mask as well as a $56 \times 56$ feature map from FPN-P2. Experimental results in Table 5 show that the performance of binary grid refinement is worse than PatchDCT, and even DCT-Mask. This is because binary grid refinement requires the refinement module to learn 12544 ($112 \times 112$) outputs, while PatchDCT only needs to learn at most 1176 ($14 \times 14 \times 6$) outputs, which reduces the training complexity.

**Effectiveness of three-class classifier.** In addition to identifying mixed patches, a more important role of the three-class classifier is to correct previously mispredicted foreground and background patches. To validate the effectiveness of refining non-mixed patches (i.e. foreground and background patches), we construct a binary-class classifier, which only classifies patches as mixed or non-mixed and keeps masks of non-mixed patches unchanged. As shown in Table 6, the binary-class classifier is inferior to our three-class classifier by 0.3% AP and 0.4% AP*, since the refinement of previously incorrectly predicted foreground and background patches is ignored.

Refinement of foreground and background patches can also be accomplished with the DCT vector regressor. However, as discussed in Sec. 3.2, the DCT vector elements of the non-mixed patches

Table 5: Mask AP obtained by different refinement methods on *val2017*. PatchDCT significantly improves the quality of masks.

| Method | AP | $\text{AP}_B$ | $\text{AP}^*$ | $\text{AP}^*_B$ |
|---|---|---|---|---|
| Binary grid | 35.7 | 23.2 | 39.6 | 29.1 |
| Two-stage DCT | 36.6 | 23.9 | 40.1 | 29.1 |
| PatchDCT | **37.2** | **24.7** | **40.8** | **30.1** |

Table 6: Mask AP obtained by PatchDCT with two-class classifier and three-class classifier on *val2017*. PatchDCT with three-class classifier achieves the best performance.

| Classifier | AP | $\text{AP}_S$ | $\text{AP}_M$ | $\text{AP}_L$ | $\text{AP}_B$ | $\text{AP}^*$ | $\text{AP}^*_B$ |
|---|---|---|---|---|---|---|---|
| 2-class | 36.9 | 18.2 | 39.3 | 53.5 | 24.4 | 40.4 | 29.7 |
| 3-class | **37.2** | **18.3** | **39.5** | **54.2** | **24.5** | **40.8** | **30.1** |

Table 7: Mask AP obtained by PatchDCT with regressor focusing on all patches and mixed patches on *val2017*. The best results are obtained by regressing only the mixed patches.

| Regressor | AP | $\text{AP}_S$ | $\text{AP}_M$ | $\text{AP}_L$ | $\text{AP}_B$ | $\text{AP}^*$ | $\text{AP}^*_B$ |
|---|---|---|---|---|---|---|---|
| all | 36.6 | 17.7 | 39.5 | 52.2 | 23.6 | 39.6 | 28.6 |
| mixed | **37.2** | **18.3** | **39.5** | **54.2** | **24.5** | **40.8** | **30.1** |

Table 8: Mask AP obtained by PatchDCT with and without the regressor on *val2017*. PatchDCT benefits from the regressor.

| Regressor | AP | $\text{AP}_S$ | $\text{AP}_M$ | $\text{AP}_L$ | $\text{AP}_B$ | $\text{AP}^*$ | $\text{AP}^*_B$ |
|---|---|---|---|---|---|---|---|
|  | 36.7 | 18.3 | 39.0 | 53.1 | 23.3 | 39.6 | 27.1 |
| ✓ | **37.2** | **18.3** | **39.5** | **54.2** | **24.5** | **40.8** | **30.1** |

Table 9: Mask AP obtained by models with different dimensions of patch DCT vectors on COCO *val2017*. Model with 6-dimensional vectors achieves the best performance.

| Patch Dim. | AP | $\text{AP}_S$ | $\text{AP}_M$ | $\text{AP}_L$ | $\text{AP}_B$ | $\text{AP}^*$ | $\text{AP}^*_B$ |
|---|---|---|---|---|---|---|---|
| 3 | 36.8 | 17.6 | 39.2 | 53.5 | 24.0 | 40.5 | 29.5 |
| 6 | **37.2** | **18.3** | **39.5** | **54.1** | **24.5** | **40.8** | **30.1** |
| 9 | 36.9 | 17.1 | 39.3 | 53.3 | 24.3 | 40.6 | 30.1 |

Table 10: Mask AP obtained by multi-stage PatchDCT on *val2017*. Two-stage PatchDCT achieves a trade-off between accuracy and computational complexity.

| Stage | AP | $\text{AP}_S$ | $\text{AP}_M$ | $\text{AP}_L$ | $\text{AP}_B$ | $\text{AP}^*$ | (G)FLOPs | FPS |
|---|---|---|---|---|---|---|---|---|
| 1 | 37.2 | **18.3** | 39.5 | 54.1 | 24.5 | 40.8 | **5.1** | 12.3 |
| 2 | **37.4** | 17.8 | **40.0** | 54.0 | **24.7** | **41.2** | 9.6 | 11.1 |
| 3 | 37.3 | 17.3 | 39.7 | **54.6** | **24.7** | 40.9 | 14.1 | 8.4 |

Table 11: Mask AP obtained by models with different patch sizes on COCO *val2017*. PatchDCT with $8 \times 8$ patch size obtains the best performance.

| Patch Size | AP | $\text{AP}_S$ | $\text{AP}_M$ | $\text{AP}_L$ | $\text{AP}_B$ | $\text{AP}^*$ | $\text{AP}^*_B$ |
|---|---|---|---|---|---|---|---|
| $4 \times 4$ | 37.0 | 17.5 | 39.3 | 53.8 | 24.4 | 40.5 | 29.8 |
| $8 \times 8$ | **37.2** | **18.3** | **39.5** | **54.1** | **24.5** | **40.8** | **30.1** |
| $16 \times 16$ | 37.0 | 17.6 | 39.3 | 53.5 | 24.4 | **40.8** | 30.0 |

Table 12: Mask AP obtained by models with different feature map sizes on COCO *val2017*. The performance saturates with the $42 \times 42$ feature map.

| Feature Size | AP | $\text{AP}_S$ | $\text{AP}_M$ | $\text{AP}_L$ | $\text{AP}_B$ | $\text{AP}^*$ | $\text{AP}^*_B$ |
|---|---|---|---|---|---|---|---|
| $28 \times 28$ | 37.1 | 17.8 | 39.3 | 53.4 | 24.5 | 40.6 | 30.0 |
| $42 \times 42$ | **37.2** | **18.3** | **39.5** | **54.1** | **24.5** | 40.8 | **30.1** |
| $56 \times 56$ | 37.0 | 17.4 | 39.2 | 53.0 | 24.4 | **41.0** | 30.3 |

Table 13: Mask AP obtained by PatchDCT with the feature map cropped from all levels and P2 only on COCO *val2017*. Model with the feature map of P2 obtains higher mAP.

| Feature | AP | $\text{AP}_S$ | $\text{AP}_M$ | $\text{AP}_L$ | $\text{AP}_B$ | $\text{AP}^*$ | $\text{AP}^*_B$ |
|---|---|---|---|---|---|---|---|
| P2 | **37.2** | **18.3** | **39.5** | **54.1** | **24.5** | **40.8** | **30.1** |
| P2-P5 | 37.1 | 18.2 | 39.3 | 53.3 | 24.4 | 40.6 | 29.8 |

only involve zero and $m$, making it ineffective to learn the DCT vectors of all patches directly. As shown in Table 7, the performance of the method refining non-mixed regions with the DCT vector regressor is lower than the method using a three-class classifier by 0.6% AP and 1.2% $\text{AP}^*$. Need to note that, $\text{AP}_B$ and $\text{AP}^*_B$ decrease by 0.9% and 1.5% respectively, reflecting that learning to regress non-mixed patches also affects the prediction of boundaries.

**Effectiveness of the regressor.** The regressor is actually a boundary attention module that generates finer boundaries. As shown in Table 8, after removing the regressor and keeping only the classifier, the overall AP only decreases by 0.5% , but $\text{AP}_B$ and $\text{AP}^*_B$ decrease by 1.2% and 3.0% respectively. The phenomenon demonstrates the importance of the regressor for generating finer boundaries.

**Dimension of PatchDCT vectors** We look for an appropriate patch DCT vector length to encode each mixed patch. Results in Table 9 show that the model with 6-dimensional patch DCT vectors obtains the best performance. As also shown in Table 1, the 6-dimensional patch DCT vector already contains most of the ground truth information. As more elements bring only very little incremental information, regressing these elements does not improve the prediction.

**Multi-stage PatchDCT.** We compare the performance of the multi-stage procedure in Table 10. One-stage PatchDCT already provides high-quality masks, while two-stage PatchDCT further improves the prediction. However, the computational cost of the mask branch has nearly doubled with tiny improvements in the quality of masks, so we choose to use one-stage PatchDCT in our paper.

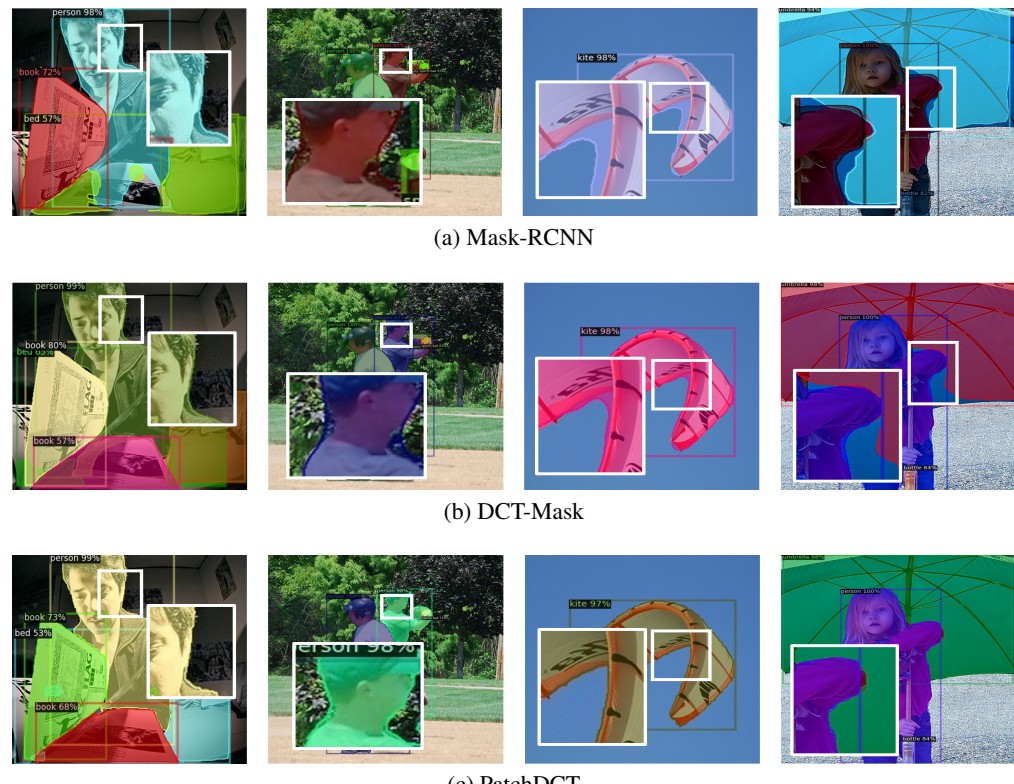

(a) Mask-RCNN

(b) DCT-Mask

(c) PatchDCT

Figure 4: COCO example tuples from Mask-RCNN, DCT-Mask, and PatchDCT. Mask-RCNN, DCT-Mask and PatchDCT are trained based on R50-FPN. PatchDCT provides masks with higher quality and finer boundaries.

**Size of the patch.** We evaluate the influence of patch size in Table 11. We keep the resolution of the mask and the size of the input feature map unchanged and compare the model performance with different patch sizes. PatchDCT with $8 \times 8$ patches performs better than other settings.

**Size of the feature map.** We compare the model with different sizes of the feature map used in PatchDCT. Table 12 illustrates that the performance saturates with the $42 \times 42$ feature map.

**Feature map from FPN.** We evaluate PatchDCT with the feature map cropped from all pyramid levels or P2. Table 13 shows that PatchDCT benefits from the finer feature map of P2.

### 4.5 QUALITATIVE RESULTS

In Figure 4 we visualize some outputs of PatchDCT on COCO *val2017*. PatchDCT generates finer boundaries among different instances, such as the shoulder of the person (the first column), the contour of the kite (the third column), and the arm of the girl (the fourth column). PatchDCT obtains masks of higher quality in comparison with Mask-RCNN and DCT-Mask.

## 5 CONCLUSIONS

In this work, we propose PatchDCT, a compressed vector based method towards high-quality instance segmentation. In contrast to previous methods, PatchDCT refines each patch of masks respectively and utilizes patch DCT vectors to compress boundaries that are full of details. By using a classifier to refine foreground and background patches, and predicting an informative low-dimensional DCT vector for each mixed patch, PatchDCT generates a high-resolution mask with fine boundaries. PatchDCT is designed with a simple and clean structure, which allows the method to obtain high-quality segmentation with almost negligible cost in speed compared to Mask-RCNN and DCT-Mask. We hope that our approach will benefit future studies in instance segmentation.

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

## A  MORE QUALITATIVE RESULTS

### A.1  TWO-STAGE DCT

We visualize some outputs of two-stage DCT and compare them with DCT-Mask to demonstrate the disadvantages of simply combining DCT-Mask with multi-stage progress.

As shown in Figure 5, in two-stage DCT, the areas that were previously correctly predicted may be influenced in refinement. The phenomenon further proves the difficulties in refining DCT vectors directly.

### A.2  QUALITATIVE RESULTS ON CITYSCAPES

We show some qualitative results on Cityscapes in Figure 6. In comparison with Mask-RCNN and DCT-Mask, PatchDCT generates finer boundaries that greatly improve the quality of masks.

## B  MORE TECHNICAL DETAILS

We prove that all elements except the DCCs for foreground patches are zero.

It can be derived from Equation 6 that DCC is equal to the patch size $m$ in the foreground patch since $M_{m \times m}(x, y) = 1$.

$$DCC = \frac{1}{m} \sum_{x=0}^{m-1} \sum_{y=0}^{m-1} M_{m \times m}(x, y) = m,  \tag{6}$$

Note that for a $m \times m$ patch $M_{m \times m}^{f}(u, v)$ Equation 1 can be written as

$$M_{m \times m}^{f}(u, v) = \frac{2}{m} C(u)C(v) \left( \sum_{x=0}^{m-1} A(x, u) \right) \left( \sum_{y=0}^{m-1} A(y, v) \right),  \tag{7}$$

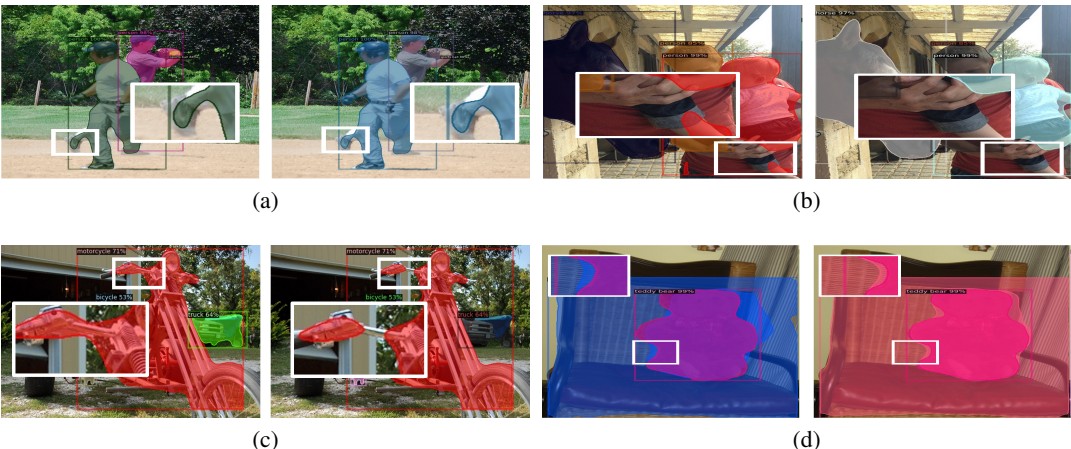

Figure 5: Visualization of DCT-Mask (left) and two-stage DCT (right). Areas that were correctly predicted are influenced by the refinement.

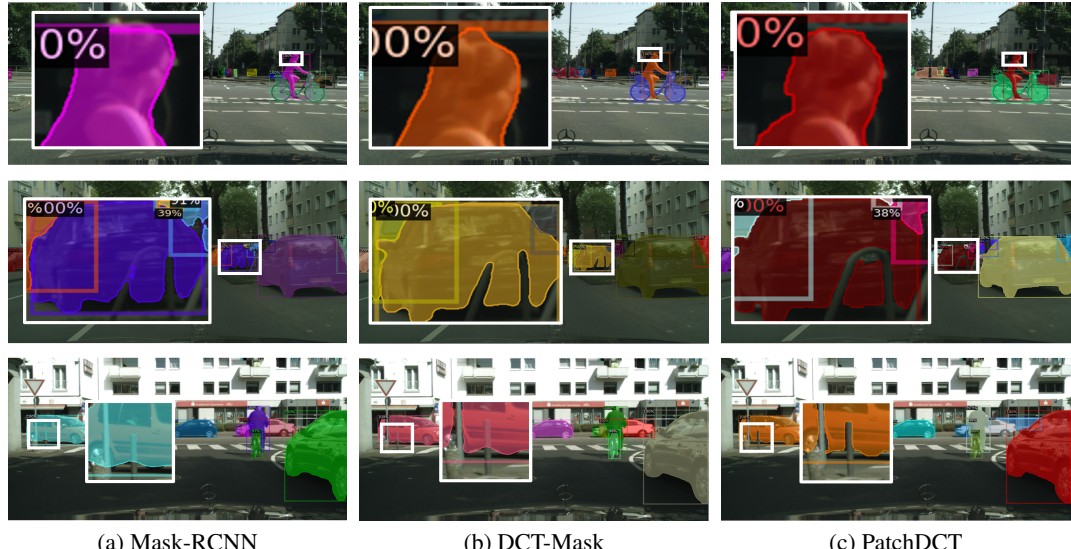

(a) Mask-RCNN      (b) DCT-Mask      (c) PatchDCT

Figure 6: Cityscapes example tuples from Mask-RCNN, DCT-Mask, and PatchDCT. Mask-RCNN, DCT-Mask and PatchDCT are trained based on R50-FPN. PatchDCT generates masks with finer boundaries.

where $A(a, b) = cos\frac{(2a+1)b\pi}{2m}$.

If $u$ is odd,

$$
\begin{aligned}
A(m - 1 - x, u) &= cos\frac{(2(m - 1 - x) + 1)u\pi}{2m} \\
&= cos\left(-\frac{(2x + 1)u\pi}{2m} + u\pi\right) \\
&= -A(x, u),
\end{aligned}
\tag{8}
$$

If $u$ is even and larger than zero, since from Euler's formula

$$
e^{i\theta} = cos\theta + isin\theta,
\tag{9}
$$

We have

$$\sum_{x=0}^{m-1} A(x,u) = \sum_{x=0}^{m-1} cos\frac{(2x+1)u\pi}{2m}$$

$$= Re\left(\sum_{x=0}^{m-1} e^{\frac{(2x+1)u\pi i}{2m}}\right)$$

$$= Re\left(e^{\frac{u\pi i}{2m}}\frac{1-e^{u\pi i}}{1-e^{\frac{u\pi i}{m}}}\right) = 0, \tag{10}$$

Since $u$ is even,

$$e^{u\pi i} = cos(u\pi) + isin(u\pi) = 1, \tag{11}$$

We obtain

$$\sum_{x=0}^{m-1} A(x,u) = 0, \quad \forall u \neq 0, \tag{12}$$

Therefore for foreground patches

$$M_{m\times m}^f(i,j) = \left\{ \begin{array}{ll} m, & i=0, j=0, \\ 0, & otherwise. \end{array} \right. \tag{13}$$

This illustrates except the DCCs, elements of DCT vectors of foreground patches are all zero.

## C  LIMITATIONS AND FUTURE OUTLOOK

In the process of visualization, we observe that the model may generate masks with holes. These problems usually occur in semantical ambiguous areas, and rarely in the center of the mask where the semantic information is very clear. We demonstrate some typical bad cases in Figure 7. In these cases, the model either misclassifies these patches or generates imprecise patch DCT vectors, resulting in disconnected masks. We leave better classification and regression vectors as future work. In addition, we also plan to carry out further verification in other more challenging areas, such as aerial images, medical images, etc. Taking aerial images as an example, this field still focuses on the research of object detection (Yang et al., 2019; 2021a;b;c; 2023), especially oriented object detection (Yang & Yan, 2022; Zhou et al., 2022; Yang et al., 2022), which lacks the exploration of more precise positioning tasks, i.e instance segmentation.

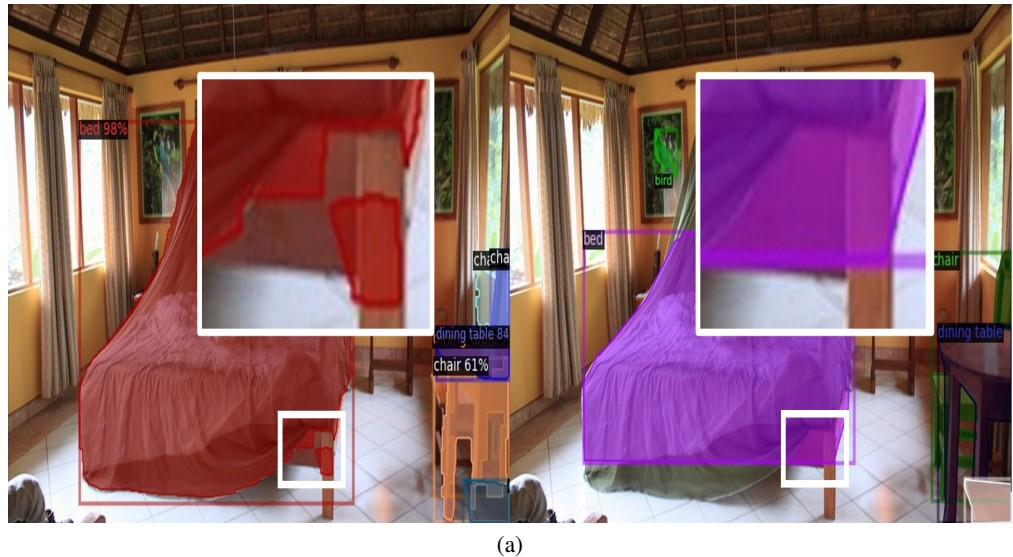

(a)

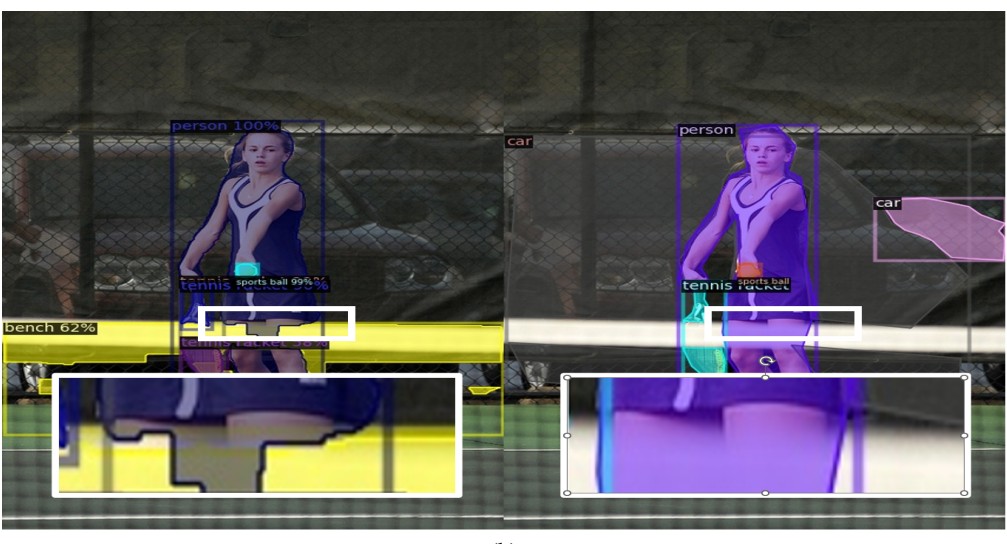

(b)

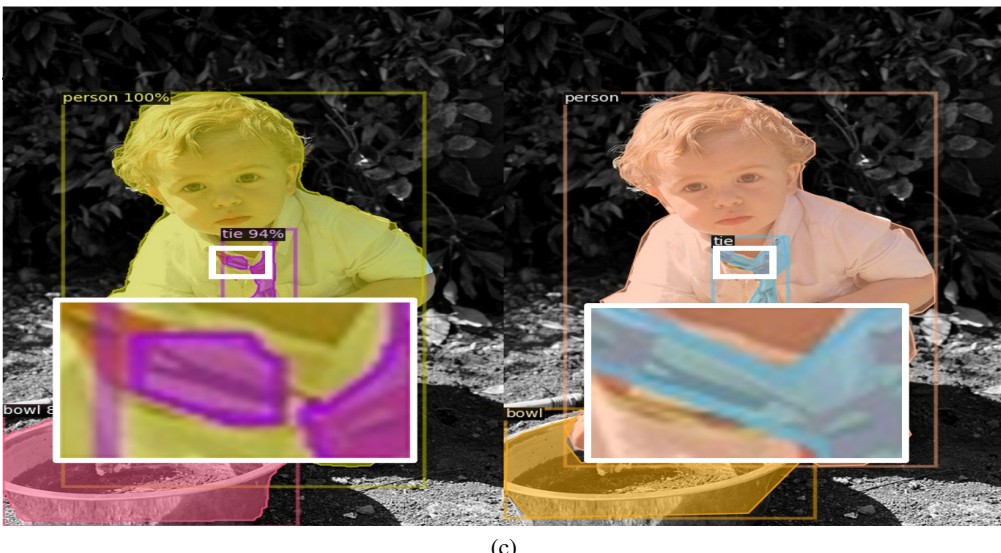

(c)

Figure 7: Visualization of typical bad cases of our model, PatchDCT (left) and ground truth (right).

