# OpenReview forum: "PatchDCT: Patch Refinement for High Quality Instance Segmentation"
_ICLR.cc/2023/Conference — ICLR 2023 poster_

### Official Review · Reviewer_FMns · 2022-10-23

**Confidence:** 4
**Correctness:** 3
**Technical Novelty And Significance:** 3
**Empirical Novelty And Significance:** 3
**Recommendation:** 8

**Clarity, Quality, Novelty And Reproducibility:**

Clarity:
In general, it is clear. However, in the 4.3 section, the second paragraph did not clearly give reference to its data points. Further, it mentioned that "MaskTransifer runs at 5.5 FPS on the A100 GPU, which is almost two times slower than PatchDCT", however, the paper did not provide concrete data points.

Quality:
In general, the paper is well-written.

Novelty:
The reviewer believes that the novelty of this paper is good.

**Strength And Weaknesses:**

Strengths:
1. Introducing the patching technique to refine the generated masks to improve boundary segmentation performance and this idea is of good interest.
2. As patching is the key technique in the paper, the patching size is also analyzed by experiments to provide a suggested 8*8 size for users. Also, other hyperparameters like the dimension of PatchDCT vectors and the number of stages for PatchDCT are clearly given and discussed.
3. Many experiments are done to support the design.

Weaknesses:
1. Backbone is limited to CNN-based models. Vision transformer-based model which also uses a patching technique would be of interest to see whether they will make any difference to the conclusions.

**Summary Of The Paper:**

The paper adds a patching technique to the DCT-Mask model and adopts a refinement technique for each patch, so that high-resolution masks can be achieved. The patching technique can produce better boundaries compared to the DCT mask model itself as element changes for DCT vectors can be limited to the patch level rather than the entire mask. The paper also provides many experiments to verify the capability of the proposed method and compare it with state-of-the-art methods. Ablation experiments are also provided to discuss the efficacy of the designed framework.

**Summary Of The Review:**

In sum, the reviewer believes that the paper provides a new refinement method to improve the results of the DCT-mask model, and experiments are done to support the design of its framework. However, it is recommended that the author address the clarity issue mentioned above, and the following comments:
1. In table 4, with the same backbone R101-FPN, SOLQ seems to have a better performance than PatchDCT with R101-FPN. However, the authors did not give any clarification in the paper.
2. For the classifier given in the paper in Figure 2, if a foreground patch is assigned to the background, will this patch make a rectangle hole in the object?
3. If $N$ and $n$ are the same notation, please make them consistent.

---------------------------------------------------------------------------------------------------------
Updated: 11/27
I appreciate that the authors well-addressed my comments and questions. I raise the score to 8.

---

> ### Author Response · Authors · 2022-11-10
> **Response to Reviewer FMns (Round 1)**
>
> > ***Q1:  Backbone is limited to CNN-based models. Vision transformer-based model which also uses a patching technique would be of interest to see whether they will make any difference to the conclusions.***
>
> **A1** We evaluate the performance of PatchDCT with Swin-B backbone on COCO 2017 test-dev. We have updated the result in Table 4 (marked in red), which is higher than Mask Transfiner by 0.7% AP. The detailed results are as follows:
> |Method| Backbone|MS|Sched. |AP|FPS|
> |----------------|-------------------------------|-----------------------------|-----------------------------|-----------------------------|-----------------------------|
> MaskTransfiner|Swin-B|&check;|3x|45.9|3.5|
> PatchDCT|Swin-B|&check;|3x|46.6|7.3|
> > ***Q2: In table 4, with the same backbone R101-FPN, SOLQ seems to have a better performance than PatchDCT with R101-FPN. However, the authors did not give any clarification in the paper.***
>
> **A2:** SOLQ requires 50 epochs for training. PatchDCT with R101-FPN uses the 3x setting in detectron2, which is about 36 epochs. Even with less training time, PatchDCT is only 0.2% lower than SOLQ, but 1.1 fps faster than it.
>
> > ***Q3:  For the classifier given in the paper in Figure 2, if a foreground patch is assigned to the background, will this patch make a rectangle hole in the object?***
>
> **A3:** Thanks for your insight comment. As you said, we do find that PatchDCT may produce masks with holes. This issue usually occur in semantical ambiguous areas, and rarely in the center of the mask where the semantic information is very clear. We update some typical bad cases in the Appendix (Figure 7). In these cases, the model either misclassifies the patches or generates imprecise patch DCT vectors, resulting in disconnected masks.
>
>
> > ***Q4: If  $N$  and  $n$  are the same notation, please make them consistent.***
>
> **A4:** $N$ is the dimension of the DCT vector of the entire mask and $n$ is the dimension of patch DCT vectors. We set $N=300$ and $n=6$ in the paper. Therefore, $N$ and $n$ are not the same notation.
>
> > ***Q5:  About clarity: mentioned that "MaskTransifer runs at 5.5 FPS on the A100 GPU, which is almost two times slower than PatchDCT", however, the paper did not provide concrete data points.***
>
> **A5:** In the latest version, we have updated the speed comparison with other SOTA methods and add supplemental description (refer to the red part in Table 4 and Section 4.3). Runtime is measured on a single A100. We present a brief version of Table 4 below to demonstrate the excellent trade-off between performance and speed.
> |Method| Backbone|MS|Sched. |AP|FPS|
> |----------------|-------------------------------|-----------------------------|-----------------------------|-----------------------------|-----------------------------|
> Mask RCNN|R101-FPN|&check;|3x|38.8|13.8
> DCT-Mask|R101-FPN|&check;|3x|40.1|13.0
> MaskTransfiner|R101-FPN|&check;|3x|40.7|5.5
> SOLQ|R101-FPN|&check;|50e|40.9|10.7
> HTC|RX101-FPN||20e|41.2|4.3
> PointRend|RX101-FPN|&check;|3x|41.4|8.4
> RefineMask|RX101-FPN|&check;|3x|41.8|8.9
> PatchDCT|R101-FPN|&check;|3x|40.7|11.8
> PatchDCT|RX101-FPN|&check;|3x|42.2|11.7

---

### Official Review · Reviewer_K4c6 · 2022-10-23

**Confidence:** 4
**Clarity, Quality, Novelty And Reproducibility:** The writing is good and presentation …
**Correctness:** 3
**Technical Novelty And Significance:** 2
**Empirical Novelty And Significance:** 2
**Recommendation:** 5

**Strength And Weaknesses:**

The paper introduces PatchDCT, which improves the quality of instance segmentation. The experiments show the competitive performance of the proposed method. The paper provides detailed information for reproduction.
There some previous works also focus on the segmentation boundary, such as Fully Connected CRF in DeepLab[1], CRFasRNN[2]. Comparison to these methods maybe helpful.

[1] DeepLab: Semantic Image Segmentation with Deep Convolutional Nets, Atrous Convolution, and Fully Connected CRFs

[2] Conditional Random Fields as Recurrent Neural Networks

**Summary Of The Paper:**

This paper is focused on instance segmentation. Baseline method applise Discrete Cosine Transform to improve the segmentation quality around the object boundary. This paper extends DCT-Mask from image-level to patch-level. Through Figure 4 we could see the quality improvement visually.

**Summary Of The Review:**

The main concern is the absence of the comparision to the previous work, therefore I mark it as bordline. I am happy to change my rating if the authors can address this  question.

---

> ### Author Response · Authors · 2022-11-10
> **Response to Reviewer K4c6 (Round 1)**
>
> > ***Q1: There some previous works also focus on the segmentation boundary, such as Fully Connected CRF in DeepLab[1], CRFasRNN[2]. Comparison to these methods maybe helpful.***
>
> **A1:** Thanks for your suggestion. We have added these works into the Related Work of the latest version, and also compare them with PatchDCT.

---

> ### Author Response · Authors · 2022-11-18
> **Response to Reviewer K4c6 (Round 1 Update)**
>
> In the latest version, we update the comparison between Panoptic-DeepLab and PatchDCT on Cityscapes val (refer to the red part in Table 3).
> The detailed results are as follows:
> |Method| AP|AP$_{50}$|AP$_B$|
> |----------------|-------------------------------|-----------------------------|-----------------------------|
> Mask-RCNN|33.7|60.9|11.8|
> Panoptic-Deeplab|35.3|57.9|16.5|
> PointRender|35.9|61.8|16.7|
> DCT-Mask|36.9|62.9|14.6|
> RefineMask|37.6|63.3|17.4|
> Mask Transfiner|37.9|64.1|18.0|
> PatchDCT|38.2|64.5|18.8|

---

> > ### Comment · Reviewer_K4c6 · 2022-11-21
> > **Thanks for the reply and I keep initial score**
> >
> > The main contribution of this method is improving the segmentation quality around the object boundary. we have CRF-based method to get a more accurate boundary in previous years, therefore I think a direct comparison of refining module is necessary. However, the authors only added an end-to-end result with different experimental setting. After reading the Table, I cannot figure out whether this DCT-based module is better than CRF-based module or not. Therefore I keep the initial score.

---

> ### Author Response · Authors · 2022-11-21
> **Response to Reviewer K4c6 (Round 2)**
>
> Thank you for your reply. Your initial provided works [1-2] are all about semantic segmentation, so we are not sure whether your more accurate boundary refers to instance segmentation task, and whether the fairness of the settings is maintained during comparison. Can you provide some references about CRF based instance segmentation, because we haven't collected any CRF based work that is better than ours?
>
> As far as we  know, CRF and PatchDCT are two different technologies. CRF is often used in semantic segmentation as a post-processing operation (not end-to-end, time-consuming, parameter sensitive) to refine the entire prediction results, not just for the boundary. In contrast, PatchDCT proposes a learnable network structure to efficiently refine the boundary in an end-to-end manner for instance segmentation. Therefore, is it meaningful to compare the two methods? We hope the reviewer can give further advice.
>
> In this paper, we have made a more in-depth study on the DCT based method (compared with the published DCT Mask), not to say that the DCT based method must be better than CRF based. We also think that this comparison is meaningless for the time being, because the development of technology is diversified and there is no absolute advanced method.

---

> > ### Author Response · Authors · 2022-11-22
> > **Supplement on the relationship between CRF and PatchDCT**
> >
> > PatchDCT and CRF are in different parts of the detector, i.e. the former is in the head of the detector, and the latter is in the post-processing stage. In other words, they cannot replace each other. Theoretically, these two technologies can be used at the same time. The following table shows our experimental results after careful hyperparameter tuning:
> >
> > Table. All models use R50-FPN backbone. AP is measured on COCO 2017val.
> > |                                                                             Method|Resolution| CRF | AP | AP$_S$ | AP$_M$ | AP$_L$ |AP$_B$|FPS|
> > |:------------------------------------------------------------------------------------ |:----------------:|:--------------:|:---------------:|:----------:|:-----------:|:----------:|:-----------:|:-----------:|
> > Mask-RCNN|28x28||**35.2**|**17.2** |**37.7** |**50.3**|**21.1**|**13.9**|
> > |||&check;|34.7|16.9|37.1|49.5|20.0|13.1|
> > DCT-Mask|128x128||36.5| 17.7| 38.6| **51.9**| **23.6**|**13.2**|
> > |||&check;|36.5|**17.8**|38.6|51.7|23.0|3.5|
> > PatchDCT|112x112||**37.2** |18.3 |**39.5**| **54.2**|**24.5**| **12.3**|
> > |||&check;|37.1|18.3|39.4|53.9|24.3|3.4
> >
> > We use open source code [1] for CRF experiments. We can draw the following conclusions:
> >
> > 1. CRF and the proposed method (PatchDCT)  can be used simultaneously and work on different parts of the detector. Their correlation is weak and direct comparison is not appropriate.
> >
> > 2. CRF post-processing does not improve the performance of instance segmentation, but causes a serious decline in speed especially for high resolution based methods. This may be the main reason why many advanced instance segmentation methods do not use CRF (as far as we know).
> >
> > 3. Compared with CRF, our proposed method can significantly improve the performance of instance segmentation, and has no significant speed drop. In this respect, our method is superior to CRF.
> >
> > [1] https://github.com/lucasb-eyer/pydensecrf

---

### Official Review · Reviewer_feWT · 2022-10-24

**Confidence:** 4
**Correctness:** 4
**Technical Novelty And Significance:** 3
**Empirical Novelty And Significance:** 3
**Recommendation:** 8

**Clarity, Quality, Novelty And Reproducibility:**

For one thing, the work starts with the issue of straightforward refining the global DCT vectors, proposes a patch-based method to solve this issue and finally gets well performance. It's logically fluent and complete. The experiment result of multiple metrics on three popular instance-segmentation datasets is also clear.

For another, as a refinement method, the work clearly shows that very small-size compressed vectors (such as 6-dimension) can afford enough information for segmentation. This is a valuable result for the refinement task.

**Strength And Weaknesses:**

Strengths :

1. The paper clearly identifies that straightforward refining the global DCT vetors is unsuitable and proposes a patch-based method to overcome this issue.
2. Foreground patch and background patch have their special DCT vector, refiner them with a three-classes-classifier rather than a general DCT regressor is theoretically sound.
3. Compared with many other methods, this work achieves SOTA results, and ablation studies are sufficient.

Weaknesses :
1. There is only runtime result compared with Mask-RCNN and DCT-Mask. Please complement more experiments to compare the efficiency of PatchDCT with other refinement models.
2. In this paper, the result in Table 1 suggests that when using 1x1 patch and 1-dim DCT vector the network has the best performance (57.6 AP). But when encoding 1x1 patch (single-pixel) using DCT, the result should be the value of the pixel itself. What is the difference between this method and directly refining the mask with 1x1 conv when the patch size is 1x1? I think this result is inconsistent with DCT-Mask, nor "binary grid refinement". According to DCT-Mask (Table 1), directly increasing the resolution decreases the mask AP, which is the main reason they use DCT encoding.

**Summary Of The Paper:**

The paper proposes a DCT-vector-based multi-stage refinement framework named PatchDCT, which contains a classifier and a regressor. PatchDCT first separates the original coarse mask into several patches. The classifier is used to distinguish mixed patches which consist of both foreground and background pixels. Then, these mixed patches are refined by the regressor with DCT-vector representation.


**Summary Of The Review:**

The compressed vector-based refinement method is relatively novel. The experiment results of multiple metrics clearly show good performance in segmentation accuracy. Complementing more experiments to compare the efficiency with other refinement models could determine further superiority of the method.

---

> ### Author Response · Authors · 2022-11-10
> **Response to Reviewer feWT (Round 1)**
>
> > ***Q1:   There is only runtime result compared with Mask-RCNN and DCT-Mask. Please complement more experiments to compare the efficiency of PatchDCT with other refinement models.***
>
> **A1:** In the latest version we update the speed comparison with other SOTA methods and add supplemental description (refer to the red part in Table 4 and Section 4.3). Runtime is measured on a single A100. We present a brief version of Table 4 below to demonstrate the excellent trade-off between performance and speed.
> |Method| Backbone|MS|Sched. |AP|FPS
> |----------------|-------------------------------|-----------------------------|-----------------------------|-----------------------------|-----------------------------|
> Mask RCNN|R101-FPN|&check;|3x|38.8|13.8
> DCT-Mask|R101-FPN|&check;|3x|40.1|13.0
> MaskTransfiner|R101-FPN|&check;|3x|40.7|5.5
> SOLQ|R101-FPN|&check;|50e|40.9|10.7
> HTC|RX101-FPN||20e|41.2|4.3
> PointRend|RX101-FPN|&check;|3x|41.4|8.4
> RefineMask|RX101-FPN|&check;|3x|41.8|8.9
> PatchDCT|R101-FPN|&check;|3x|40.7|11.8
> PatchDCT|RX101-FPN|&check;|3x|42.2|11.7
>
>
> > ***Q2: In this paper, the result in Table 1 suggests that when using 1x1 patch and 1-dim DCT vector the network has the best performance (57.6 AP). But when encoding 1x1 patch (single-pixel) using DCT, the result should be the value of the pixel itself. What is the difference between this method and directly refining the mask with 1x1 conv when the patch size is 1x1? I think this result is inconsistent with DCT-Mask, nor "binary grid refinement". According to DCT-Mask (Table 1), directly increasing the resolution decreases the mask AP, which is the main reason they use DCT encoding.***
>
> **A2:** Table 1 shows the evaluation results of the joint **ground-truth mask** and the bbox predicted by Mask-RCNN, while Table 1 in DCT-Mask shows results obtained by **predicted masks** of Mask RCNN. The results in Table 1 of PatchDCT are actually **the upper bound of mask AP** that the Mask-RCNN can achieve with the same bbox prediction capability. When patch size = 1 and dim = 1, there is no compression process and no ground-truth information is lost, so the setting has the highest upper bound of mask AP, i.e. 57.6 AP. Because of the slight loss of ground truth information, the upper bound decreases  marginally (mask AP from 57.6 to 57.1) when patch size=8 and dim=6.
>
> The binary grid refinement described in Section 4.4 is actually the case of patch size=1 and dim=1. However, as analyzed in Section 4.4, simply refining 112x112 masks with the binary grid representation has 12544 (112 × 112 x 1) outputs to predict, while PatchDCT only needs to learn at most 1176 (14 × 14 × 6) outputs, which eases the training process and achieved better mask AP. This is consistent with DCT-Mask, which reduces 16384(128x128) predictions to 300 and obtains performance gain.

---

### Official Review · Reviewer_NpMa · 2022-10-24

**Confidence:** 5
**Correctness:** 4
**Technical Novelty And Significance:** 3
**Empirical Novelty And Significance:** 3
**Recommendation:** 8

**Clarity, Quality, Novelty And Reproducibility:**

Clear writing and good paper quality; extensive experiments with effective improvement.

**Strength And Weaknesses:**

Strength:
1. Figure 1 shows clear motivation of the method. Although inspired by previous methods (such as PointRend and Mask Transfiner), dividing whole image to three classes of patches and refine the mixed patch in a multi-stage is a good strategy.
2. Extensive ablation experiments comparison, and state-of-the-art method results (although improvement is limited).
3. Results improvement comparing to DCT-mask with also small speed decrease.
4. Good paper writing and clear structure, which is easy for readers to understand.

Weakness:
1.  Can the paper describes more on the speed advantages compared to previous SOTA methods? What's the speed of using one-stage PatchDCT and two-stage PatchDCT respectively?
2. What are the typical failure/bad cases of the proposed methods?

**Summary Of The Paper:**

The paper proposes PatchDCT for high-quality instance segmentation. Different from DCT-Mask, the whole image mask is divided into different patches. Each patch is refined individually by the classifier and regressor. The refinement is performed in a multi-stage. Improvements on mask quality are observed on COCO, Cityscapes and LVIS.

**Summary Of The Review:**

The effective improvement validated by experiments; interesting idea of dividing patches into 3 classes and refining each one with classifier and regressor; clear motivation.

---

> ### Author Response · Authors · 2022-11-10
> **Response to Reviewer NpMa (Round 1)**
>
> > ***Q1: Can the paper describes more on the speed advantages compared to previous SOTA methods? What's the speed of using one-stage PatchDCT and two-stage PatchDCT respectively?***
>
> **A1.1** Thanks for your comment. In the latest version we update the speed comparison with other SOTA methods and add supplemental description (refer to the red part in Table 4 and Section 4.3). Runtime is measured on a single A100. We present a brief version of Table 4 below to demonstrate the excellent trade-off between performance and speed.
> |Method| Backbone|MS|Sched. |AP|FPS
> |----------------|-------------------------------|-----------------------------|-----------------------------|-----------------------------|-----------------------------|
> Mask RCNN|R101-FPN|&check;|3x|38.8|13.8
> DCT-Mask|R101-FPN|&check;|3x|40.1|13.0
> MaskTransfiner|R101-FPN|&check;|3x|40.7|5.5
> SOLQ|R101-FPN|&check;|50e|40.9|10.7
> HTC|RX101-FPN||20e|41.2|4.3
> PointRend|RX101-FPN|&check;|3x|41.4|8.4
> RefineMask|RX101-FPN|&check;|3x|41.8|8.9
> PatchDCT|R101-FPN|&check;|3x|40.7|11.8
> PatchDCT|RX101-FPN|&check;|3x|42.2|11.7
>
>
> **A1.2:** The speeds using one-stage PatchDCT and two-stage PatchDCT with R50-FPN are shown in the Table below:
> |Method|AP|AP$^*$|(G)FLOPs|FPS
> |----------------|-------------------------------|-----------------------------|-----------------------------|-----------------------------|
> |one-stage|37.2|40.8|5.1|12.3
> |two-stage|37.4|41.2|9.6|11.1
>
> We also update the speed of PachDCT for different stages in Table 10 (refer to the red part). We observe that although two-stage PatchDCT achieves a certain improvement over one-stage PatchDCT, the computational cost increases and the inference speed reduces. For the trade-off between performance and computational cost, we use one-stage PatchDCT in the paper.
>
> > ***Q2: What are the typical failure/bad cases of the proposed methods?***
>
> **A2:** We observe that misclassification or imprecise patch DCT vectors regression may result in masks with holes. These problems usually occur in semantical ambiguous areas. We update in the Appendix some typical bad cases. In these cases, the model either misclassifies the patches or generates imprecise patch DCT vectors, resulting in disconnected masks (refer to Section C in Appendix).

---

### Author Response · Authors · 2022-11-18
**Comments to all Reviewers**

Dear Reviewers,

Approaching the pdf updating ddl, is there anything needing added.

Best,

Paper267 Authors

---

### Decision · Program_Chairs · 2023-01-20

**Decision:**

Accept: poster

**Justification For Why Not Higher Score:**

In terms of a 'surprise factor' that I think is important on whether a paper should be recommended as a spotlight or an oral, I think the paper is lacking. The main idea is close to what other papers have proposed (DCT for segmentation), but a clever combination and/or application improves results further. All in all, I think the paper brings a solid contribution and a poster presentation is sufficient.

**Justification For Why Not Lower Score:**

See metareview.

**Metareview: Summary, Strengths And Weaknesses:**

In a gist, the paper proposes mainly to use Discrete Cosine Transforms at a patch and not just image level, so that improve segmentation quality around boundaries.

This paper received strong acceptance scores from 3 out of 4 reviewers. They all acknowledge the idea is novel enough, the experiments convincingly contribute to the literature, and the writing is clear so that to be positively accepted by the community.

There was criticism on whether the positioning and comparisons with related work is sufficient. The authors include theoretical comparison in the related work, they also show that the method is similarly efficient with Mask-RCNN and DCT-Mask in terms of FPS, while using same backbone architectures.

All in all, under scrutiny, this paper was well received, and I recommend acceptance.

**Note From Pc:**

if the above contains the word "oral" or "spotlight" please see: "oral" presentation means -> notable-top-5% and "spotlight" means -> notable-top-25%. As stated in our emails, we are disassociating presentation type from AC recommendations